# SARS-CoV-2 transmission and impacts of unvaccinated-only screening in populations of mixed vaccination status

Kate M. Bubar[1,6], Casey E. Middleton[2,6], Kristen K. Bjorkman ⓘ [3], Roy Parker ⓘ [3,4,5] & Daniel B. Larremore ⓘ [2,3✉]

Screening programs that test only the unvaccinated population have been proposed and implemented to mitigate SARS-CoV-2 spread, implicitly assuming that the unvaccinated population drives transmission. To evaluate this premise and quantify the impact of unvaccinated-only screening programs, we introduce a model for SARS-CoV-2 transmission through which we explore a range of transmission rates, vaccine effectiveness scenarios, rates of prior infection, and screening programs. We find that, as vaccination rates increase, the proportion of transmission driven by the unvaccinated population decreases, such that most community spread is driven by vaccine-breakthrough infections once vaccine coverage exceeds 55% (omicron) or 80% (delta), points which shift lower as vaccine effectiveness wanes. Thus, we show that as vaccination rates increase, the transmission reductions associated with unvaccinated-only screening decline, identifying three distinct categories of impact on infections and hospitalizations. More broadly, these results demonstrate that effective unvaccinated-only screening depends on population immunity, vaccination rates, and variant.

---

[1] Department of Applied Mathematics, University of Colorado Boulder, Boulder, CO, USA. [2] Department of Computer Science, University of Colorado Boulder, Boulder, CO, USA. [3] BioFrontiers Institute, University of Colorado Boulder, Boulder, CO, USA. [4] Department of Biochemistry, University of Colorado Boulder, Boulder, CO, USA. [5] Howard Hughes Medical Institute, Chevy Chase, MD, USA. [6] These authors contributed equally: Kate M. Bubar, Casey E. Middleton. ✉email: daniel.larremore@colorado.edu

SARS-CoV-2 has created a pandemic in which morbidity and mortality have been partially mitigated in many areas by widespread vaccination. COVID-19 vaccines have been extremely effective at preventing severe disease (vaccine efficacy, VE > 90%[1]), while also reducing susceptibility to infection ($VE_S$) and risk of onward transmission ($VE_I$). In spite of these reductions, so-called vaccine breakthrough infections and subsequent transmission have been widely documented[2], and have increased dramatically with the emergence of the omicron variant in late 2021[3,4]. These developments raise the question of how to best mitigate transmission in partially vaccinated populations.

Prior to the approval of COVID-19 vaccines, transmission mitigation via regular and repeated screening was shown to be an effective approach to break chains of transmission and decrease the burden of COVID-19 using both RT-PCR[5–7] and rapid antigen testing[7,8]. Specifically, screening via testing, which we hereafter refer to as simply screening in most cases, is effective at the community level because it decreases transmission from individuals who are already infected[7,9]. However, policy proposals in 2021 and early 2022 shifted to focus routine testing requirements on only the unvaccinated population, including an Italian requirement announced in October, 2021[10] and a U.S. requirement for healthcare workers beginning February, 2022[11]. By reducing rates of transmission from only the unvaccinated population, such policies may be limited by the extent to which transmission is, in fact, driven by the unvaccinated. This raises critical questions about projected policy impacts relative to other non-pharmaceutical interventions (NPIs)[12,13]—particularly in areas where the unvaccinated population is small.

The role of vaccines in reducing transmission is complex and changing. First, $VE_S$ and $VE_I$ vary depending on which vaccine was administered[14]. Second, both $VE_S$ and $VE_I$ wane with time since vaccination[15–17], but can be boosted to higher levels for those receiving an additional dose[18]. Third, those who have experienced a SARS-CoV-2 infection also show decreased risks of reinfection and subsequent transmission[14], providing partial protection to those who are previously infected and remain unvaccinated, but also augmenting protection for those who are both vaccinated and previously infected[18,19]. Finally, preliminary estimates of $VE_S$ and $VE_I$, and their prior infection equivalents, are markedly lower for the omicron variant[3,20]. Thus, the relative estimates of risk reductions due to vaccination, prior infection, or both, as well as the sizes of the populations falling into each category of immunity, will affect transmission dynamics—with or without testing.

In this study, we model the spread of SARS-CoV-2 in populations of mixed vaccination status, focusing on three critical questions. First, how do vaccinated and unvaccinated populations each contribute to community spread and hospitalizations, and how do those contributions vary with rates of vaccination and prior infection? Second, how do testing-based screening programs focused on unvaccinated individuals alone affect community spread and hospitalizations? Third, how effective are delta-era screening strategies likely to be against variants with higher breakthrough and reinfection rates? Our study's goals are not to make perfectly calibrated predictions but instead to elucidate more general principles of transmission and unvaccinated-only testing in partially vaccinated populations. As such, our analyses consider a wide range of parameters and scenarios.

## Results
**Unvaccinated-only testing-based screening programs** have been discussed and implemented during transmission of both the delta and omicron variants, yet these variants differ in their transmission and disease parameters, particularly among vaccinated or

previously infected individuals—the focus of the present study. As such, the analyses that follow incorporate a range of empirically estimated parameters for the delta variant and plausible parameters associated with the omicron variant.

**High vaccination rates drive total infections and hospitalizations down, increase the proportions of vaccine breakthroughs, and shift the drivers of transmission.** To examine the dynamics of transmission in a population with mixed vaccination status, we first modeled transmission within and between communities of vaccinated (V) and unvaccinated (U) individuals in the absence of a screening program. Based on a standard susceptible exposed infected recovered (SEIR) model, we tracked the four transmission modes by which an infection might spread: $U \rightarrow U$, $U \rightarrow V$, $V \rightarrow U$, and $V \rightarrow V$ (Fig. 1a). A constant and equivalent fraction of both populations was assumed to have experienced prior SARS-CoV-2 infection, resulting in four categories of imperfect immunity: unprotected (unvaccinated with no prior infection), infection-acquired, vaccine-acquired, and both vaccine- and infection-acquired (so-called "hybrid" immunity). To account for introductions of infection from outside the population, all susceptible individuals were subject to a small, constant rate of exposure, with infection-acquired and vaccine-acquired immunity providing partial protection against subsequent infection.

Traditionally, the basic reproductive number $R_0$ is defined as the number of secondary infections generated by a typical infector in an entirely susceptible population, i.e., a population without any NPIs. At the time of writing, NPIs such as masking, ventilation and physical distancing are commonplace in many areas, so we hereafter define $R_0^{\mathrm{NPI}}$ to be the expected number of secondary infections generated by a typical infector in a population with possible NPIs, but prior to any impacts of population immunity. Furthermore, because precise estimates of $R_0^{\mathrm{NPI}}$ vary by context, variant, and over time, we consider a range of $R_0^{\mathrm{NPI}}$ values from 4 to 6. In our baseline modeling scenario, vaccines were assumed to reduce susceptibility to infection by $VE_S = 65\%$, the likelihood of transmission to others by $VE_I = 35\%$, and the likelihood of disease progression to hospitalization conditioned on infection by $VE_P = 86\%$, values which land within plausible literature estimates for the effectiveness of two doses of mRNA vaccine against the delta variant in the absence of dramatic waning and without boosting[14,18,21,22]. Though less often studied in the literature, we assumed that prior SARS-CoV-2 infection would lead to 63% and 13% decreases in risks of infection and transmission, respectively, based on a statistical model relating immunity to neutralization[18], and that hybrid immunity would be superior to either vaccination or prior SARS-CoV-2 infection alone. We further assumed an additional 54% decrease in risk of hospitalization conditioned on infection for individuals with prior SARS-CoV-2 infection[23], and that individuals with hybrid immunity receive the greater of vaccinated or prior immunity's protection against hospitalization. See Materials and Methods and Supplementary Tables 1 and 2 for a complete description of the model and parameters.

In a modeled population of $N = 20,000$ with 58% vaccination rate (corresponding to U.S. estimates as of Nov. 4, 2021[24]) and 35% past infection rate, outbreaks still occurred, despite assuming a partially mitigated delta variant ($R_0^{\mathrm{NPI}} = 4$). During the ensuing outbreak, 59% of total infections and 89% of hospitalizations occurred in unvaccinated individuals (Fig. 1b, c), despite making up only 42% of the population. Furthermore, the peak burden of disease occurred first in the unvaccinated community and then 1 week later in the vaccinated community (Fig. 1b), a known consequence of disease dynamics in populations with

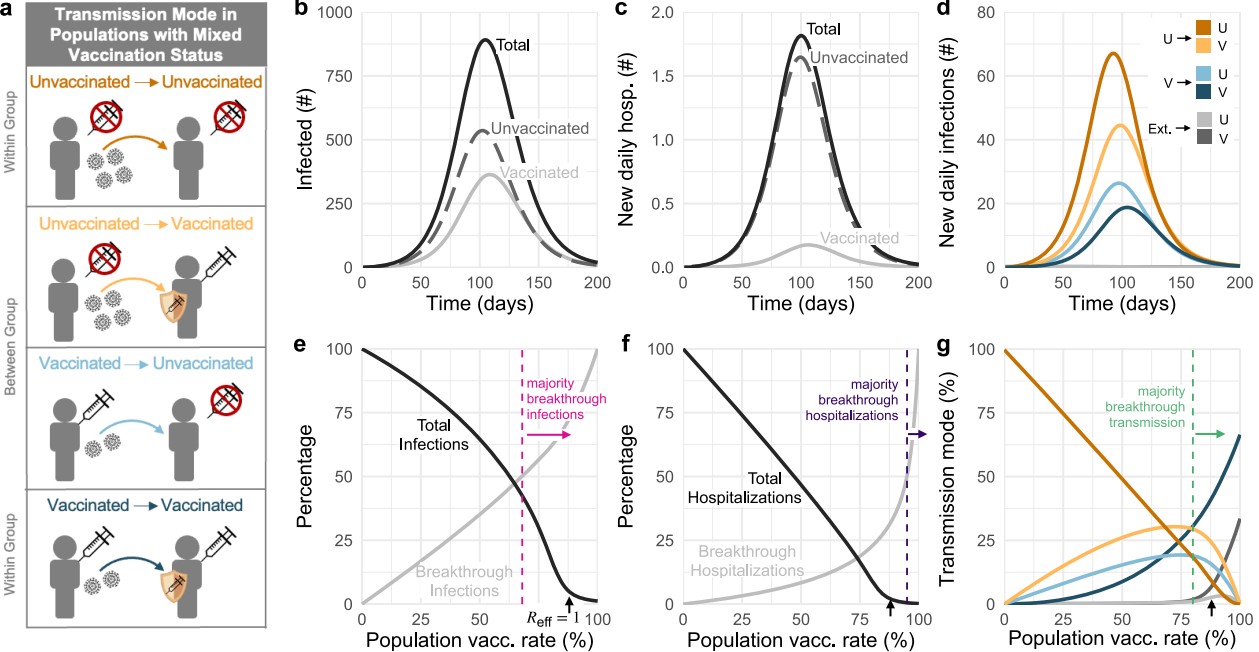

**Fig. 1 Vaccination affects which population drives transmission and dominates infections and hospitalizations. a** Diagram of four transmission modes within and between vaccinated and unvaccinated communities, where vaccines and prior infection decrease risks of infection, transmission, and hospitalization. **b** Number of infected individuals and **c** new daily hospitalizations over time (solid black), stratified by unvaccinated (dashed gray) and vaccinated (solid gray) populations. **d** Daily transmission events separated and colored by transmission mode (see legend). **e** Cumulative infections and **f** hospitalizations as a percentage of total infections/hospitalizations without vaccination (black), and the percent of each accounted for by vaccine breakthroughs (gray) for varying vaccination rates. **g** Transmission mode (see legend) as a percentage of cumulative infections for varying vaccination rates. Black arrows in **e–g** indicate vaccination rate at which $R_{eff} = 1$; vertical dashed lines indicate the lowest vaccination rates for which vaccinated individuals account for the majority of infections, hospitalizations, and transmission as annotated. $R_0^{NPI} = 4$ for all plots, with baseline VE and immunity parameters vs. the delta variant (Materials and Methods, Supplementary Tables 1 and 2); no screening. **b–d**: 58% vaccination rate and 35% rate of prior infection.

heterogeneous susceptibility and transmissibility[25,26]. By categorizing transmission events into four distinct modes (Fig. 1a), we observe that infections during a delta outbreak in both communities were driven predominantly and consistently by the unvaccinated community ($U \rightarrow U$, $U \rightarrow V$; Fig. 1d), but that there was nevertheless some transmission from the vaccinated community (breakthrough transmission). These differences occurred despite a "well-mixed" modeling assumption—namely, that an individual with a given vaccination status is no more or less likely to associate with a member of their own group vs. the other group.

Vaccination and past infection rates vary widely across the U.S.[24] and the world[27] due to impacts of both vaccine availability[27] and refusal[28], as well as the success or failure of transmission mitigation policies. We therefore asked how a population's vaccination and past infection rates would affect our observations about infections, hospitalizations, and the relative impacts of the four modes of transmission. This analysis revealed three important points.

First, our results reinforce the fact that increased vaccination rates lead to decreased total infections and hospitalizations, both before and after the herd immunity threshold at $R_{eff} = 1$ (Fig. 1e, f). Moreover, when large proportions of the population are also partially protected by immunity from prior infection, the vaccination levels required to reach $R_{eff} = 1$ decrease considerably (Fig. 2a). For instance, increasing prior infection rates from 35 to 50% decreases the required vaccination rate for $R_{eff} = 1$ from 87 to 80% under baseline modeling parameters. Combinations of immunity from past infection and vaccination thus have the potential to create a herd immunity frontier, beyond which transmission is no longer self-sustaining even

in the absence of screening. We caution that although total infections and hospitalizations may appear equal along lines of constant $R_{eff}$ (Fig. 2a, b), both actually decrease as vaccination rates increase, due to vaccines' superior protection, relative to prior infection, against infection and hospitalization.

Second, as vaccination rates increased, the fraction of infections classified as vaccine breakthroughs increased (Fig. 1e), creating a transition point such that when 68% of the population was vaccinated, 50% of all infections were breakthrough infections under our baseline modeling conditions for the delta variant. To determine whether this transition point of 68% was sensitive to the precise fraction of the population with immunity from past infection (35%, Fig. 1), we varied the fraction with infection-acquired immunity between 0 and 100%, finding that the 50/50 breakthrough infection transition occurred between 63 and 75% vaccine coverage (Fig. 2c). Because vaccines provide an additional level of protection against hospitalization $VE_P$, the 50/50 breakthrough hospitalization transition occurs at rates of vaccination of 90–96% (Figs. 1f and 2e). Thus, our results set the expectation that increasing vaccination rates will decrease total infections and hospitalizations, yet a higher proportion of both will be breakthroughs, irrespective of levels of immunity due to prior infection.

Third, as vaccination rates increased, the unvaccinated community ceased to be the primary driver of transmission. Under our baseline modeling conditions ($R_0^{NPI} = 4$, 35% with infection-acquired immunity), this transition occurred when 80% or more of the population was vaccinated (Fig. 1g). When we varied the fraction of the population with infection-acquired immunity between 0 and 100%, this transition point varied from

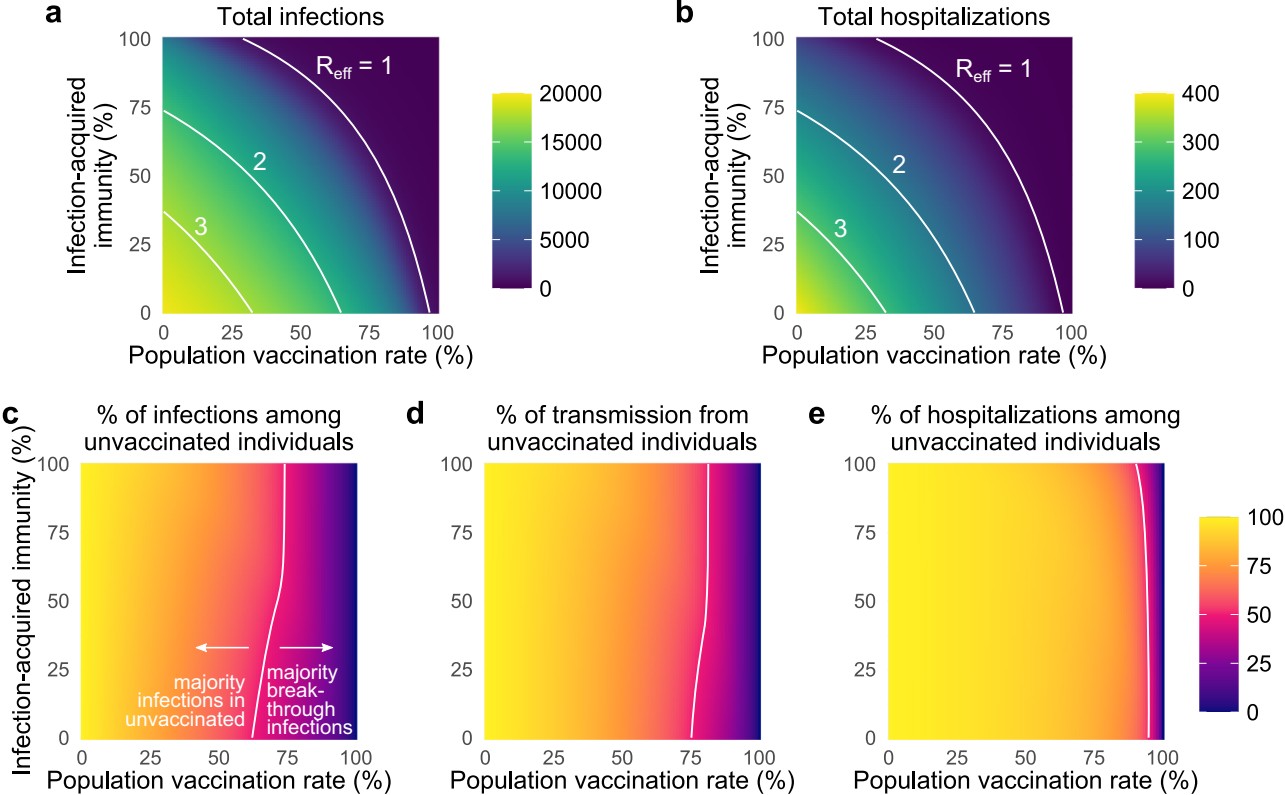

**Fig. 2 Vaccination and prior infection rates affect epidemic potential, vaccine breakthroughs, and drivers of transmission.** Heatmaps show **a** the total number of infections, **b** the total number of hospitalizations, **c** the percentage of total infections occurring in the unvaccinated population, **d** the percentage of total infections caused by the unvaccinated population, and **e** the percentage of total hospitalizations occurring in the unvaccinated population for simulated epidemics (see text). White annotation curves show (**a**, **b**) isoclines of the effective reproductive number $R_{\mathrm{eff}}$ calculated at $t = 0$, and the line of parameters along which (**c**) 50% of infections were breakthroughs, **d** 50% of transmission was due to breakthrough infections, and **e** 50% of hospitalizations were breakthroughs. $N = 20{,}000$ and $R_0^{\mathrm{NPI}} = 4$ for all plots, with baseline VE and immunity parameters vs. the delta variant (Materials and Methods, Supplementary Tables 1 and 2); no testing. See Supplementary Fig. 2 for $R_0^{\mathrm{NPI}} = 6$.

76 to 82% (Fig. 2d). Thus, while COVID-19 hospitalizations remain concentrated primarily in unvaccinated populations, only a minority of infections will occur in, and be driven by, the unvaccinated community when vaccine coverage is sufficiently high. Note that this implies that unvaccinated individuals living in highly vaccinated communities will still be exposed to SARS-CoV-2 and thus remain at risk of infection and severe disease.

These findings are driven by reductions in susceptibility, disease severity, and infectiousness arising from vaccination, prior SARS-CoV-2 infection, or both. However, quantitative estimates of those reductions vary depending on which vaccine was administered[17], time since vaccination or SARS-CoV-2 infection[15–17], whether an additional "booster" dose was given[18], and the variant circulating at the time of the study[29,30]. We therefore sought to determine how our findings might change under different sets of assumptions about vaccine effectiveness by comparing our baseline scenario ($VE_S = 0.65$, $VE_I = 0.35$, $VE_P = 0.86$) with a waning/low immunity scenario ($VE_S = 0.5$, $VE_I = 0.1$, $VE_P = 0.80$) and a boosted/high immunity scenario ($VE_S = 0.8$, $VE_I = 0.6$, $VE_P = 0.90$), as well as a scenario reflecting plausible VE values based on early observations for the omicron variant ($VE_S = 0.35$, $VE_I = 0.05$, $VE_P = 0.77$[22,31]). To explore the impact of these changes in vaccine effectiveness, we simulated outbreaks for all combinations of vaccination and infection-acquired immunity rates under the four VE scenarios. Across simulations, total infections and hospitalizations were well predicted by calculating $R_{\mathrm{eff}}$ at the start of each simulation (Eq. (3); Methods). In particular, outbreaks were small when vaccination or past infection rates crossed the herd immunity threshold ($R_{\mathrm{eff}} < 1$).

When $R_{\mathrm{eff}} > 1$, total infections monotonically increased as $R_{\mathrm{eff}}$ increased (Supplementary Fig. 1). The herd immunity threshold was impossible to cross with vaccination alone in the waning or omicron VE scenarios with partially mitigated transmission ($R_0^{\mathrm{NPI}} = 4$, Fig. 3a, d and Supplementary Fig. 1), and in waning, baseline, and omicron VE scenarios with unmitigated transmission ($R_0^{\mathrm{NPI}} = 6$; Supplementary Fig. 1), as evidenced by the fact that the $R_{\mathrm{eff}} = 1$ curves either fail to intersect the vaccination rate axis or appear at all.

Waning, boosting, or omicron-specific assumptions altered the proportions of infections and hospitalizations occurring in, and transmission from, the unvaccinated vs. vaccinated communities. All else being equal, waning or omicron VE led to increased fractions of breakthrough infections and hospitalizations, and increased transmission from the vaccinated community, while boosted VE led to decreases of all three. In turn, the population vaccination rates at which the majority of infections or hospitalizations were breakthroughs shifted down for waning or omicron VE (Fig. 3a, d), while the vaccination rate at which the majority of transmission was driven by vaccinated individuals shifted up for boosted VE (Fig. 3c).

Among the four transition points identified in transmission dynamics, we observe that, in each VE scenario, $R_{\mathrm{eff}}$ is driven by both vaccination and past infection rates, as evidenced by curvature in $R_{\mathrm{eff}} = 1$ isoclines (Fig. 3, black lines). In contrast, isoclines representing the transition points between majority-unvaccinated vs. majority-breakthrough infections (Fig. 3, pink lines), between

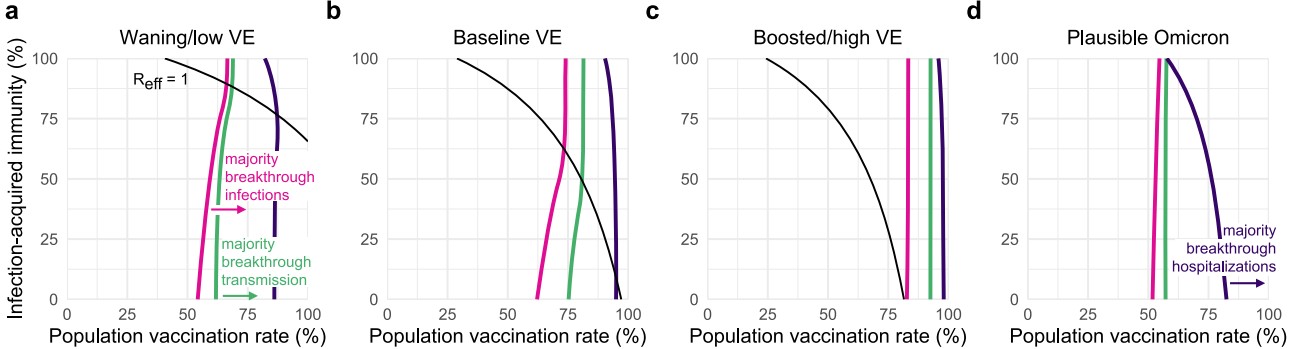

**Fig. 3 Transition points for breakthrough infections, hospitalizations, and transmission.** All panels show curves representing the vaccination and prior infection rates infections (pink), transmission (green), and hospitalizations (purple) are composed of equal numbers of vaccinated an unvaccinated individuals, with majority-breakthrough regions to the right of each line as indicated, for **a** waning/low, **b** baseline, and **c** boosted/high VE vs. the delta variant, and **d** plausible VE vs. the omicron variant. Black lines indicate $R_{eff} = 1$ isoclines, which do not appear in **d** due to $R_{eff} > 1$. See Supplementary Table 2 for immunity parameter values. $R_0^{NPI} = 4$ in all panels.

majority-unvaccinated vs. majority-breakthrough hospitalizations (Fig. 3, purple lines), and between majority-unvaccinated vs. majority-breakthrough transmission (Fig. 3, green lines) are relatively insensitive to variation in rates of infection-acquired immunity, as evidenced by vertical or near-vertical isoclines. These findings suggest that the relative proportions of breakthrough infections, hospitalizations, and transmission are driven more by vaccination rates and VE, but not by rates of past infection or proximity to herd immunity; indeed, after the herd immunity threshold, all three isoclines show essentially no variation. These observations suggest that unvaccinated-only screening programs, which decrease rates of $U \rightarrow U$ and $U \rightarrow V$ transmission, may be highly effective only in regimes where transmission is driven by the unvaccinated (i.e., to the "left" of green isoclines, Fig. 3), an intuition we now explore in detail.

**The impacts of unvaccinated-only screening depend on population immunity, compliance, and VE.** To explore the impact of unvaccinated-only screening on population transmission, we modified our simulations so that a positive test would result in an unvaccinated individual isolating to avoid infecting others[7,32]. We considered test sensitivity equivalent to RT-PCR with a 1-day delay between sample collection and diagnosis under three screening paradigms: weekly testing with 50% compliance—a value which reflects observed compliance with a weekly testing mandate in a university setting[5]—weekly testing with 99% compliance, and, specifically for the omicron variant, twice-weekly testing with 99% compliance.

Our analysis shows that the benefits of an unvaccinated-only screening program fall into one of three categories, depending on the population vaccination rate and transmission dynamics. These categories align with three distinct regions in parameter space, denoted in Fig. 4 as regions I, II and III. In region I, screening is insufficient to fully control transmission, yet nevertheless markedly reduces the peak number of total infections, colloquially "flattening the curve" (Fig. 4a). In region II, screening successfully brings transmission under control (Fig. 4b). In region III, screening has little impact on transmission due to the fact that outbreaks are already mitigated by population immunity and other control measures (Fig. 4c). Unvaccinated-only screening is therefore impactful in the first two regions, sufficient for transmission control in only the second region, and largely inconsequential to transmission in the third.

The three regions that correspond to different impacts of screening on transmission are separated by boundaries which can be estimated from two analytical calculations of $R_{eff}$—one which

includes the effects of screening and one which does not (Eq. (3), Methods). The boundary separating regions I and II is given by those parameters for which $R_{eff} = 1$ with screening, while the boundary separating regions II and III is given by those parameters for which $R_{eff} = 1$ without screening (Fig. 4d). Thus, the value of a screening testing program in reducing infections can be evaluated based on which of three regions the current vaccination rate, prior infection rate, and VE fall into.

To illustrate the value of this $R_{eff}$-based analysis, we considered vaccination rates and prior infection rates ranging from 0–100% and varied VE between waning, baseline, and boosted scenarios for the delta variant. Across scenarios, dramatic relative reductions in cumulative infections are concentrated within the envelope between the boundaries of $R_{eff} = 1$ with and without screening, i.e., region II (Fig. 5). Outside of this effective screening envelope, percent reductions in cumulative infections decreased markedly, either because unvaccinated-only screening flattened the infection curve but had little impact on cumulative infections (region I), or because existing population immunity prevented large outbreaks in the first place (region III). Assuming a 35% past infection rate and $R_0^{NPI} = 4$, region III appeared only for baseline and boosted vaccine effectiveness assumptions, and only when vaccination rates were ~90% or greater (baseline VE) or 75% or greater (boosted VE). Sensitivity analyses show that increasing $R_0^{NPI}$ to 6, potentially representing pre-pandemic contact rates and the SARS-CoV-2 delta variant, cause region III to shrink further (Supplementary Fig. 5). Thus continued screening for SARS-CoV-2 among the unvaccinated may be of limited value when vaccination rates are sufficiently high.

The role of compliance—the fraction of scheduled tests that are actually taken—can also be clarified by examining the three regions of screening testing impact. Both the simulations and equations for $R_{eff}$ show that increasing compliance from 50% (Fig. 5, row 1) to 99% (Fig. 5, row 2) causes the lower boundary of the effective screening envelope to shift to lower vaccination and prior infection rates, decreasing the size of region I and increasing the size of region II. Moreover, increased compliance increases the magnitude of infection reductions within both regions, visible as an intensification of color in the infection reduction heatmaps (Fig. 5). As a result of these observations, we conclude that, in addition to test sensitivity, frequency, and turnaround time[7], high participation in screening programs is critical to expanding the impact of unvaccinated-only screening testing programs. However, we also note that compliance had little effect in region III where $R_{eff} < 1$, a result which parallels analysis of universal screening programs[9].

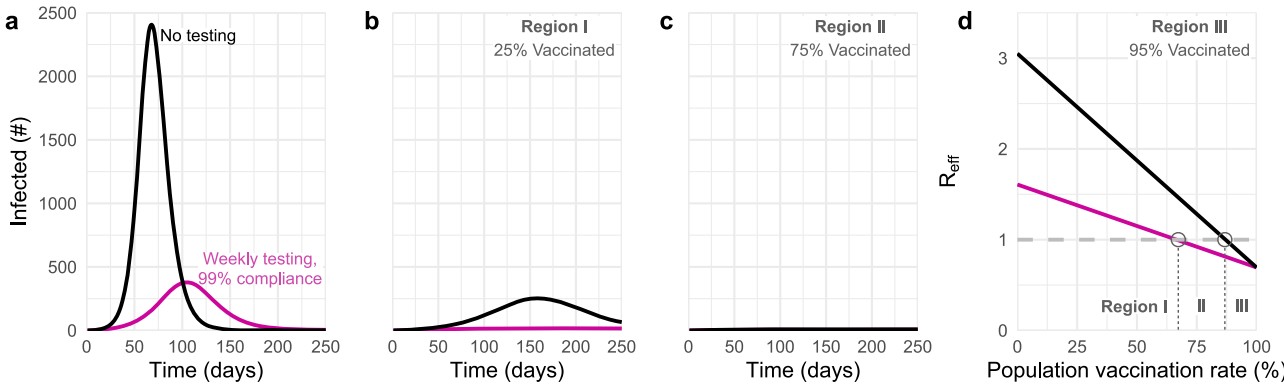

**Fig. 4 The impact of unvaccinated-only screening corresponds to three distinct parameter regions.** Total number of infections with no screening (black) and weekly testing with 99% compliance (pink) are shown for **a** 25%, **b** 75%, and **c** 95% population vaccination rates. **d** Effective reproductive number over various population vaccination rates, where $R_{eff} = 1$ is denoted by gray dashed line. The impacts of screening fall into three categories (see text) depending on whether vaccination rate falls into region I, II, or III, as annotated. $R_0^{NPI} = 4$ and 35% rate of prior infection with baseline immunity parameters (Materials and Methods, Supplementary Table 2).

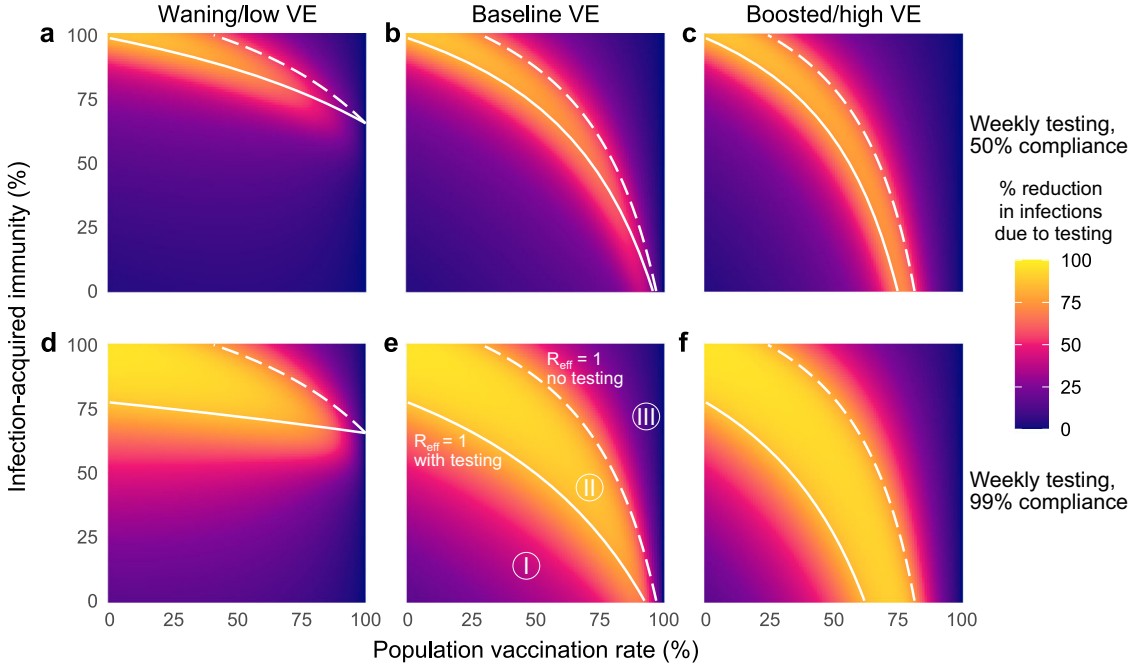

**Fig. 5 The impacts of unvaccinated-only screening depend on population immunity, compliance, and vaccine effectiveness.** Percent reduction in cumulative infections due to screening over various population vaccination rates assuming waning/low (**a**, **d**), baseline (**b**, **e**), and boosted/high (**c**, **f**) VE with once-weekly screening at 50% (top row) and 99% (bottom row) compliance. White lines indicate the population immunity rate at which $R_{eff} = 1$ with screening (solid) and without screening (dashed), which divide the space into three regions, labeled I, II and III. See Supplementary Tables 1 and 2 for parameter values. $R_0^{NPI} = 4$ in all panels; see Fig. 3 for $R_0^{NPI} = 6$.

This $R_{eff}$-based analysis of transmission is not restricted to unvaccinated-only screening programs. To illustrate this, we considered an identical set of simulations as in Fig. 5 but with universal screening, i.e., screening via testing of the vaccinated and unvaccinated populations alike. Universal screening caused the boundary between regions I and II ($R_{eff} = 1$ with screening) to shift, expanding the size of the effective screening envelope (Supplementary Fig. 6). While we present these results here for completeness, universal testing in mixed vaccination status populations have been investigated elsewhere prior to the present work[9].

The impact of screening on hospitalizations is also predicted well by the $R_{eff}$-based effective screening envelope. While hospitalizations were not identical across all equal-$R_{eff}$ combinations of vaccination and prior infection rates, dramatic

relative reductions in cumulative hospitalizations were nevertheless clearly concentrated within region II, with decreasing relative reductions in regions I and III (Supplementary Fig. 6). We therefore find that analysis based only on the effective screening envelope, calculated via Eq. (3) (Methods), is useful in predicting the impact of screening regimens—both unvaccinated-only and universal—on reductions in cumulative infections and hospitalizations alike.

The omicron variant's rapid spread, and in particular its increased rates of reinfection and vaccine breakthrough, raise key questions about the role of unvaccinated-only screening programs and whether populations considering such policies may fall into region I, II, or III defined above. Because estimates of omicron-specific immunity parameters remain either limited or nonexistent at the time of writing, we assumed plausible lower

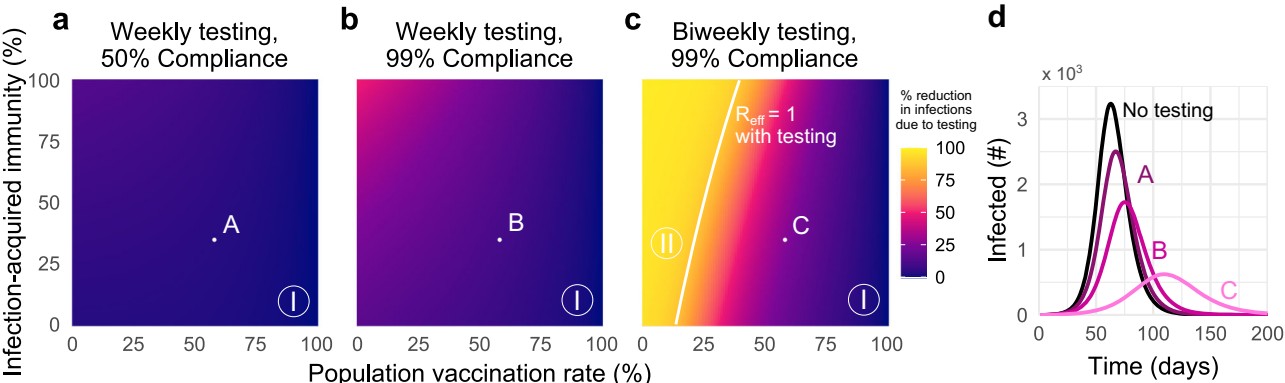

**Fig. 6 Unvaccinated-only screening during omicron transmission cannot achieve $R_{eff}$ < 1 except in low-vaccination and high-frequency regimes.** Percent reduction in cumulative infections due to screening over various population vaccination rates assuming plausible parameters for immunity vs. the omicron variant, with **a** once-weekly screening at 50% compliance, **b** once-weekly screening at 99% compliance, and **c** twice-weekly screening at 99% compliance. **d** Number of individuals infected over time, under screening scenarios denoted A, B, C, compared with no screening (black) with 58% vaccination rate and 35% rate of prior infection. Solid white line indicates $R_{eff} = 1$ with screening; $R_{eff} = 1$ is not achievable without screening. See Supplementary Tables 1 and 2 for parameter values. $R_0^{NPI} = 4$ in all panels; see Supplementary Fig. 4 for $R_0^{NPI} = 6$ and Supplementary Fig. 5 for universal testing.

values of each based on extant data[22,31], alongside lower infection hospitalization rates for omicron in general (Supplementary Tables 1 and 2). Under such assumptions, weekly unvaccinated-only screening with 50% compliance was ineffective at reducing cumulative infections even though screening reduced the peak magnitude of infections (Fig. 6d). Regardless of compliance, all prior infection and vaccination rate combinations with a weekly screening policy were in region I, indicating that the magnitude of peak infections can be mitigated but the impact on cumulative infections is low (Fig. 6a, b). Doubling the frequency of screening to twice weekly with 99% compliance creates a bifurcated landscape, with highly effective screening only in settings with 18–40% vaccination rates (Fig. 6c). For vaccination rates above 50%, even twice-weekly unvaccinated-only screening programs with near-perfect compliance are unlikely to dramatically impact the spread of the omicron variant (region I). Universal screening showed comparatively higher impact, yet, nevertheless, only twice-weekly testing regimens created broad region II regimes in which community spread was controlled (Supplementary Fig. 6).

**Unvaccinated-only screening shifts the balance of unvaccinated vs. breakthrough transmission but not infection or hospitalization.** By reducing transmission from unvaccinated individuals, screening programs specifically mitigate $U \rightarrow U$ and $U \rightarrow V$ transmission modes, thus diminishing the role of the unvaccinated population in transmission dynamics and amplifying the relative role of vaccine breakthrough transmission. As a consequence, we observe that in the presence of screening, the vaccination rates at which the unvaccinated cease to drive a majority of transmission decrease by up to 15 percentage points (Fig. 7b), with the largest decreases for 99% compliance and waning VE vs. delta, and the smallest decreases for 50% compliance and boosted VE vs. delta, or in all screening scenarios vs. the omicron variant. Under waning/low, baseline and omicron VE scenarios, unvaccinated-only screening programs shrink the regime in which the unvaccinated population drives outbreaks.

In contrast, unvaccinated-only screening programs had little effect on the percentage of infections or hospitalizations that were vaccine breakthroughs. Instead, majority-breakthrough regimes remained primarily dependent on vaccination rates and vaccine effectiveness (Fig. 7a, c), with transitions to majority-breakthrough infection regimes beginning at 55 to 67% vaccination rates (waning VE, delta), 63 to 75% vaccination rates (baseline VE, delta), 83 to 84% vaccination rates (boosted VE,

delta), and 50 to 55% vaccination rates (omicron). Transitions to majority-breakthrough hospitalizations occurred at 83 to 88% (waning VE, delta), 91 to 96% (baseline VE, delta), 96 to 99% vaccination rates (boosted VE, delta), and 58 to 83% vaccination rates (omicron), regardless of screening. We therefore conclude that unvaccinated-only screening programs do not markedly alter the expectations of majority-breakthrough infections or hospitalizations at high vaccination levels, particularly if VE is low or waning.

**Discussion**

In this analysis, we find that in communities with mixed vaccination status, routine SARS-CoV-2 screening programs focused only on the unvaccinated may reduce infections and hospitalizations, but in a manner dependent on two conditions. First, effective screening via testing requires high participation to be most impactful, reinforcing the need for mechanisms to encourage or enforce high participation. Second, when immunity from vaccination and past infection are high enough to curtail transmission on their own, or in concert with effective NPIs[12,13], testing the remaining unvaccinated population averts few infections and hospitalizations in both relative and absolute terms. On the other hand, when transmission due to reinfection and/or vaccine breakthrough is sufficiently high, unvaccinated-only screening will at best "flatten the curve" of infections, but cannot adequately control infections and hospitalizations except when testing twice weekly with near-perfect participation in low-vaccination communities. Once communities reach vaccination rates of ~40% or more, even twice-weekly unvaccinated-only screening, with near-perfect compliance and isolation adherence, cannot control the omicron variant. Thus, targeted unvaccinated screening programs are highly effective only in a restricted region of epidemiological parameter space, results echoed by similar work analyzing universal screening programs[9].

Key to understanding our study are three observations and findings. First, an unvaccinated-only screening program simply tests fewer and fewer individuals as vaccination rates increase. Second, the relative role of the unvaccinated population in driving transmission decreases as vaccination rates increase, regardless of vaccine effectiveness. Third, when vaccine effectiveness against infection and transmission wanes, unvaccinated-only screening programs decrease in impact. As a consequence, our work broadly suggests that unvaccinated-only screening is most beneficial when a population is undervaccinated and is close to, but

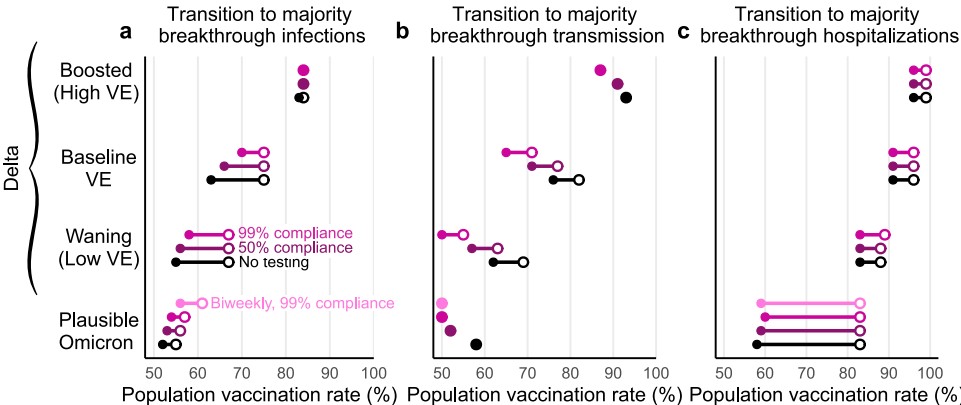

**Fig. 7 Screening and vaccine effectiveness affect transition points to majority-breakthrough regimes.** The vaccination rates at which the vaccinated population makes up the majority of **a** infections, **b** transmission, and **c** hospitalizations for low, moderate, and high vaccine effectiveness scenarios. Minimum (filled circle) and maximum (open circle) endpoints show the variation in transition points over all combinations of vaccination and prior infection rates for no screening (black), 50% compliance (purple), 99% compliance (pink) over all possible values for past infection rates. $R_0^{NPI} = 4$ for all plots; see Supplementary Fig. 7 for $R_0^{NPI} = 6$.

has not yet achieved, herd immunity—region II in our analyses—leading to the recommendation that such testing programs be used in conjunction with other NPIs. Indeed, our work finds that unvaccinated-only screening alone is generally insufficient to markedly reduce infections and hospitalizations when (1) the population is far from the herd immunity threshold, inclusive of existing NPIs, in either direction, (2) vaccination rates are high, or (3) testing is weekly and/or compliance is low. Against a backdrop of waning immunity and continued emergence of antigenically distinct variants, even herd immunity may be, at best, temporary. In this context, weekly unvaccinated-only screening programs alone are an insufficient countermeasure for the omicron variant.

Indeed, while our analysis focused on a single screening-based intervention in isolation, unvaccinated-only (or universal) screening programs are typically implemented alongside other NPIs[5,6,8]. These NPIs, including limitations on gatherings, increasing the availability of personal protective equipment, and school or restaurant closures, were estimated to have reduced the effective reproductive number $R_{eff}$ by 0.1–0.2 in 2020, particularly when implemented early[12], and by around 10% in 2021[13]. In comparison, an unvaccinated-only weekly screening policy with a realistic (50%) compliance rate[5] and 58% vaccination rate, would reduce $R_{eff}$ by an estimated 10.2% (Eq. (3)), decreasing further as vaccination rates increase, and compliance and isolation adherence decrease. Thus, while our analysis ranks vaccinate-or-test policies as potentially competitive with high-impact NPIs[12,13], such screening will decrease in impact as vaccination rates inch higher. Because prior work has shown that pandemic countermeasures also vary in their impact depending on time, vaccination, and the presence of other NPIs or behaviors[13], an empirical assessment of vaccinate-or-test programs would be valuable. However, just as empirical estimates of NPIs' impacts on $R_{eff}$ include wide uncertainty[12,13], similar estimates for unvaccinated-only screening programs are also likely to be highly uncertain.

Our study elucidates three critical transitions as vaccination rates increase. First, when vaccination rates are sufficiently high, a majority of the albeit reduced number of infections will be vaccine breakthrough infections. This fact should come as no surprise, as this transition must occur at some point for any vaccine below 100% effectiveness; for the delta variant, our modeling estimates it to take place between 63 and 75% vaccine coverage (baseline VE; 55–67% vaccinated with waning VE; 83–84% vaccinated with boosted VE), while for the omicron variant, we

estimate it to take place between 55 and 59% vaccine coverage. Second, a transition to majority-breakthrough hospitalizations will occur at some point greater than the transition to majority-breakthrough infections, a natural consequence of $VE_P > 0$. Third, while community spread is driven by the unvaccinated at low-vaccination rates, it is driven by the vaccinated population at high vaccination rates (Fig. 3). These vaccination rate transition points separating majority-unvaccinated transmission and majority-breakthrough transmission are driven lower by unvaccinated-only screening programs (Fig. 7). Taken together, these results suggest that while the overall number of infections during an outbreak decreases as vaccination rates increase, assuming $VE_S > 0$, vaccine breakthrough infections and transmission events from vaccinated individuals should not be surprising in highly vaccinated populations—vaccine effectiveness is imperfect. Consequently, in anticipation of continued community transmission even in highly vaccinated communities, those at increased risk of severe COVID-19 should take additional precautions to limit their risk of infection or severe disease.

Our analyses identify two limitations of screening via testing programs in reducing community transmission which generalize beyond the specific scenarios investigated herein. First, the ability of a screening program to prevent community spread is restricted to a limited "envelope" of past infection rate and vaccination rate combinations such that $R_{eff}$ without screening is greater than one, and $R_{eff}$ with screening is less than one. Second, the width of that effective screening envelope, and thus the effectiveness of a screening program to control transmission more broadly, are highly sensitive to compliance and participation: weekly screening with 50% compliance—a rate which reflects observed compliance in a population with a weekly screening mandate[5]—is likely to be relatively ineffective. However, although our analyses focus on the benefits of testing in reducing transmission, testing also plays an important role in diagnosis and treatment, detection of variants, situational awareness and surveillance, and decreasing pressure on the healthcare system during outbreaks. Furthermore, testing focused on the unvaccinated population may provide additional incentives to get vaccinated and thus avoid regular testing. Our study did not explore the benefits of unvaccinated-only testing mandates for these additional purposes.

Our analysis is limited in at least three different manners. First, our modeling incorporated fixed parameters that are difficult to estimate in practice. For instance, while our analysis considered boosted, baseline, and waning scenarios for vaccines' reductions

in susceptibility $VE_S$, infectiousness $VE_I$, and hospitalization given infection $VE_P$ based on ranges of estimates in the current literature, few studies are available to guide estimates of similar risk reductions associated with prior SARS-CoV-2 infection, with or without vaccination (but see refs. [14,18]). Moreover, real-time estimates for emerging variants of concern such as omicron require observational study and are thus unavailable for prospective policy analyses. Alternative parameter assumptions may be explored via the provided open source code. Second, we assumed perfect isolation after receiving a positive test result. Were this assumption to be violated by imperfect or delayed isolation, we predict a proportional loss of screening impact across all scenarios. Third, our model assumes values of $R_0^{NPI}$ and immunity associated with the delta variant and plausible values for the omicron variant, but other emerging variants may dramatically shift the values of these parameters. These limitations affect the exact vaccination and past infection rates at which the three transitions identified in our study occur, and thus our analyses describe fundamental phenomena but do not make projections or predictions for specific communities.

Our analysis also uses a well-mixed SEIR model framework, inheriting two key limitations which merit direct discussion. First, we assume that vaccination and past infection statuses are uncorrelated at the population level. In reality, they may be anticorrelated due to the protective effects of vaccination, or because those with past infection may choose to forgo subsequent vaccination. We similarly assumed no homophily in contact patterns based on vaccination status, following the well-mixed assumption of the SEIR model framework, yet those who choose to be vaccinated may be more likely to be situated in a social network with others who choose to be vaccinated, and vice versa[33]. Second, compartmental SEIR models such as ours assume uniform infectiousness in the $I$ compartment, contrasting empirical observations[34] and more sophisticated models[7,32]. While our model's latent and infectious periods are well aligned with other SEIR models[9,35–37], they nevertheless lead to unrealistically long generation times. Decreasing these periods proportionally to achieve the same reproductive number while aligning more closely with generation time estimates[38] would change the time-scale across all simulations, but would not impact the cumulative metrics or dynamics discussed in our key results.

More broadly, our work is situated within a family of research which uses mathematical modeling to estimate the impact of targeted countermeasures or strategies in populations with heterogeneous susceptibility, transmissibility, and/or contact rates. Other areas of focus include the allocation of scarce personal protective equipment to reduce transmission[39], the prioritization of vaccines by subpopulation[40–42], proactive screening programs in specific workplace structures[43] or contact networks[9], immunity "passport" programs[32], or immune shielding strategies [44]. While our analyses are directed at SARS-CoV-2, this work illustrates contributes general principles to this literature by showing that screening programs focused on testing the unvaccinated may be less effective than hoped in the face of high vaccination rates, waning vaccine effectiveness, or low compliance with testing.

## Methods

**SEIR model.** Our analyses are based on a continuous time ordinary differential equation compartmental model with susceptible, exposed, infectious, and recovered (SEIR) compartments, stratified into vaccinated $V$ and unvaccinated $U$ groups. In addition to tracking infections among these two groups separately, we also tracked infections from both groups separately, enabling us to investigate four modes of transmission: from $U$ to $U$, from $U$ to $V$, from $V$ to $U$, and from $V$ to $V$. In all simulations, we used a constant total population size of $N = 20,000$ and denoted the vaccinated fraction of the population with $\phi$.

To incorporate the possibility that individuals may have experienced prior infections, we further subdivided $U$ and $V$ into SARS-CoV-2 naive and SARS-CoV-2 experienced subpopulations, such that a fraction $\psi$ of each was assumed to be previously infected and $1 - \psi$ remains naive. For notation, we denote the subpopulations of $U$ to be $u$ (unvaccinated, naive) and $x$ (unvaccinated, experienced/prior infection), and the subpopulations of $V$ to be $v$ (vaccinated, naive) and $h$ (vaccinated, experienced). We assumed that vaccination and SARS-CoV-2 experience statuses were fixed at the start of each simulation and immutable throughout, such that there was no ongoing vaccination, and individuals who were infected and recovered during each simulation were not reassigned to SARS-CoV-2 experienced status[45].

We denote the protective effects of immunity as XE, VE, HE, expressed as reductions in risk due to prior infection alone ($x$), vaccination alone ($v$), or prior infection and vaccination (i.e., so-called "hybrid" immunity; $h$), respectively. Immunity was modeled to (1) decrease the risk of infection upon exposure, (2) decrease the risk of transmission upon infection, placing our vaccine and immunity model in the broader category of leaky models[46], and (3) decrease the risk of disease progression (i.e., hospitalization) upon infection. Reductions in the risk of infection upon exposure ($XE_S$, $VE_S$, $HE_S$), reductions in the risk of transmission when infected ($XE_I$, $VE_I$, $HE_I$), and reductions in the risk of hospitalization when infected ($XE_P$, $VE_P$, $HE_P$) were parameterized separately, based on ranges of estimates from the literature. Note that $VE_H$, the reduction in risk of hospitalization due to vaccination, is more commonly reported in the literature than $VE_P$, the reduction in risk of hospitalization due to vaccination conditional on infection. So, we used the formula $VE_P = 1 - \frac{1 - VE_H}{1 - VE_S}$ to estimate values for $VE_P$. We used the same relationship to estimate $XE_P$. See Supplementary Table 2. Due to broad uncertainty in these effects over time since exposure[14,45] or vaccination[14–16], by vaccine manufacturer and schedule[17,18,47,48], by context[30,49], and by variant[18], our analyses intentionally consider a range of values. We assumed that hybrid immunity against infection, $HE_S$, and transmission, $HE_I$, would always be superior to either vaccination alone or prior infection alone, via the simple formula $HE = (1 - XE)VE + XE$ and hybrid immunity against hospitalization given infection $HE_P = \max\{VE_P, XE_P\}$.

Supplementary Fig. 8 shows a model schematic diagram for the SEIR model used in the manuscript, where solid and dashed lines denote movement and transmission between classes, respectively. In lieu of including hospitalization as a model compartment, we computed total hospitalizations in each immunity class via multiplication of total infections, variant-specific infection hospitalization rates and $(1 - RR_P)$, where $RR_P$ is the risk reduction against progression to hospitalization given infection (i.e., 0, $VE_P$, $XE_P$, or $HE_P$) (See Supplementary Tables 1 and 2).

To model a community with open boundaries, we included a uniform risk of exposure to infection from an external source at a rate of $N^{-1}$ per person per day. For instance, in a completely naive population, $S_u/N$ individuals would be infected per day. After including the protective effects of vaccination and past infection this resulted in importation of infections at per-capita rates of $(1 - VE_S)N^{-1}$, $(1 - HE_S)N^{-1}$, $(1 - XE_S)N^{-1}$, and $N^{-1}$ new infections per day in the $v$, $h$, $x$, and $u$ groups respectively.

All simulations were run for 270 days, and all individuals were initially in one of the susceptible compartments $S_u$, $S_x$, $S_v$, or $S_h$ in proportions $(1 - \phi)(1 - \psi)$, $(1 - \phi)\psi$, $\phi(1 - \psi)$, and $\phi\psi$, respectively. Model equations were solved using *lsoda* solver from the package *deSolve*, R version 4.1.0.

**Incorporation of community testing.** Screening the unvaccinated population via community testing, and subsequent isolation of those testing positive, was modeled by increasing the rate at which infected individuals were removed from the unvaccinated $I_u$ and $I_x$ compartments. Similarly, universal screening regardless of vaccination status was modeled by increasing the rate at which infected individuals were removed from all $I$ compartments, $I_u$, $I_x$, $I_v$ and $I_h$. The effectiveness of screening tests was assumed to be equal for vaccinated and unvaccinated individuals. We estimated increased rates of removal taking into account (1) the calibrated trajectories of viral loads within individual infection[50], (2) the relationship between viral load and infectiousness[7], (3) the frequency of testing, (4) the test's analytical sensitivity (i.e., limit of detection) and turn-around time[32], and (5) screening compliance and valid sample rates, i.e., the fraction of scheduled or mandated tests which actually produce a valid sample[5]. In particular, our adaptation takes a previous model[7,32] and updates viral load dynamics for the delta variant of SARS-CoV-2[51,52], the dominant variant at the time of the present analysis. To incorporate the effectiveness of screening $\theta$, we reduce the duration of infectiousness $1/\gamma$ by a factor $(1 - \theta)$. Parameter values for $\theta$ are found in Supplementary Table 1, and are based on weekly PCR testing with a 1-day turnaround, analytical limit of detection of $10^3$ RNA copies per ml sample, and compliance rates of 50% (as in ref. [5]) or 99% (as in ref. [8]). These values assume that individuals immediately and successfully isolate upon receiving a positive diagnosis. We note that estimated effects of rapid antigen tests (with higher analytical limits of detection, but zero turnaround time) are highly similar to PCR testing under the assumptions above, provided that the community testing program frequencies and compliance rates are identical[7].

**Transmission modes and forces of infection.** Inclusive of all effects introduced above, the forces of infection are given by:

$$\lambda_u = \alpha\left(\frac{I_u}{N_u}c_{u\to u} + [1 - XE_I]\frac{I_x}{N_x}c_{x\to u} + [1 - VE_I]\frac{I_v}{N_v}c_{v\to u}\right.$$
$$\left. + [1 - HE_I]\frac{I_h}{N_h}c_{h\to u}\right) + \frac{1}{N} \tag{1}$$

$$\lambda_i = \left[\alpha\left(\frac{I_u}{N_u}c_{u\to i} + [1 - XE_I]\frac{I_x}{N_x}c_{x\to i} + [1 - VE_I]\frac{I_v}{N_v}c_{v\to i}\right.\right.$$
$$\left.\left. + [1 - HE_I]\frac{I_h}{N_h}c_{h\to i}\right) + \frac{1}{N}\right][1 - (RR_S)_i], \tag{2}$$

where $i = \{x, v, h\}$, and reductions in susceptibility due to immunity are given by $(RR_S)_i = \{XE_S, VE_S, HE_S\}$, correspondingly. The parameter $\alpha$ is the probability an unvaccinated, SARS-CoV-2 naive individual is infected by an infectious contact, tuned to achieve the desired $R_0^{NPI}$, $c_{i\to j}$ is the number of times an individual in group $j$ is contacted by individuals from group $i$ per day, and $N_j$ is a convenience variable representing the number of people in subpopulation $j$.

To produce counts of how many infections were caused by each of the transmission modes $U \to U$, $U \to V$, $V \to U$, and $V \to V$, we integrated the appropriate terms of Eqs. (1) and (2) over the duration of each simulation. For instance, the cumulative number of vaccinated infections caused by the unvaccinated population is given by integrating over the forces of infection from $u$ and $x$ to $v$ and $h$:

$$U \to V = \alpha\int_0^{270}\left[\frac{I_u(t)}{N_u}\left(c_{u\to v}S_v(t)[1 - VE_S] + c_{u\to h}S_h(t)[1 - HE_S]\right)\cdots\right.$$
$$\left. + [1 - XE_I]\frac{I_x(t)}{N_x}\left(c_{x\to v}S_v(t)[1 - VE_S] + c_{x\to h}S_h(t)[1 - HE_S]\right)\right]dt$$

**Reproductive number.** The basic reproductive number $R_0$ is defined as the expected number of secondary infections created by a typical infector in an entirely susceptible population, in the absence of any NPIs. Given the variety of environments in which SARS-CoV-2 spreads, and the presence of various permanent or semi-permanent NPIs, we use $R_0^{NPI}$ in this work to denote the reproductive number in an entirely susceptible population inclusive of varying levels of now-baseline NPIs for the delta, omicron, and future variants. We consider $R_0^{NPI} = 4$ (Main and Supplementary Figures) and $R_0^{NPI} = 6$ (Supplementary Figures). For unvaccinated-only screening programs, this model's effective reproductive number is given by:

$$R_{eff} = R_0^{NPI}\left[f_u(1 - \theta) + f_x r_x(1 - \theta) + f_v r_v + f_h r_h\right], \tag{3}$$

where $f_u = (1 - \psi)(1 - \phi)$, $f_x = \psi(1 - \phi)$, $f_v = (1 - \psi)\phi$, and $f_h = \phi\psi$ represent the fractions of the population in the unvaccinated, experienced, vaccinated, and hybrid immunity groups, respectively, and $r_x = (1 - XE_I)(1 - XE_S)$, $r_v = (1 - VE_I)(1 - VE_S)$, and $r_h = (1 - HE_I)(1 - HE_S)$ are the cumulative impacts of immunity on each group. Setting the above equation equal to a constant produces isoclines shown in plots throughout the paper. The reduction in $R_{eff}$ due to screening is given by:

$$R_{noscreening} - R_{screening} = R_0^{NPI}\theta(1 - \phi)[1 - \psi(1 - r_x)], \tag{4}$$

a function linear in each of its variables which goes to zero as the vaccination rate $\phi$ approaches 1.

For universal screening programs, similar calculations yield:

$$R_{eff}^{universal} = R_0^{NPI}(1 - \theta)[f_u + f_x r_x + f_v r_v + f_h r_h], \tag{5}$$

differing from Eq. (3) only in the terms to which $(1 - \theta)$ applies. For a complete derivation of these equations, see Supplementary Materials.

**Reporting summary.** Further information on research design is available in the Nature Research Reporting Summary linked to this article.

## Data availability
Model simulation data used in this manuscript can be found in the "dataframes" folder in the open source Github repository at https://zenodo.org/badge/latestdoi/419096560[53].

## Code availability
All code is open source and provided by the authors at https://zenodo.org/badge/latestdoi/419096560[53].

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

## Acknowledgements

The authors thank Stephen Kissler, Yonatan Grad, and Michael Mina for their feedback. K.M.B. and C.E.M. were supported in part by the Interdisciplinary Quantitative Biology (IQ Biology) Ph.D. program at the BioFrontiers Institute, University of Colorado Boulder. K.M.B. was supported by the National Science Foundation Graduate Research Fellowship under Grant No. (DGE 1650115). C.E.M. and D.B.L. were supported in part by the SeroNet program of the National Cancer Institute (1U01CA261277-01). R.P. was supported by the Howard Hughes Medical Institute.

## Author contributions

K.M.B., C.E.M., R.P., and D.B.L. conceived the study. K.M.B. and C.E.M. performed computational modeling. K.M.B., C.E.M., and D.B.L. analyzed results. All authors designed the study and wrote the manuscript.

## Competing interests

R.P. is a founder of Faze Therapeutics. D.B.L. is an advisor to Darwin BioSciences and has received consulting fees from iCareDx. All other authors declare no conflicts of interest.
