## [Peer Review File · Nature Communications]

Reviewers' Comments:

Reviewer #1:

Remarks to the Author:

The manuscript "SARS-CoV-2 Transmission and Impacts of Unvaccinated-Only Testing in Populations of Mixed Vaccination Status" by Bubar et al. presents a modelling study to assess the effectiveness of testing-for-mitigation for different SARS-CoV-2 virus variants and degrees/types of immunization in the population. This is a timely and important contribution that provides valuable input to inform discussions regarding the design of testing strategies to ensure a safer transition to endemicity, as less and less people remain in the population that are immunologically naive.

The paper is well-written and presented, literature review is adequate and methods and results are described very thoroughly. I just have a couple of minor comments regarding the interpretation of the results that the authors might want to consider.

One of the key sections of the paper is in my view the second paragraph in the discussion. The authors write: "Our analysis of plausible omicron variant transmission parameters in particular (Fig. 6) strongly suggests that weekly unvaccinated-only testing is insufficient, and that universal testing programs with high frequency and compliance are required to control transmission in highly vaccinated communities. However, since omicron infections are generally not severe in vaccinated individuals, the resources required to test entire well vaccinated communities may be better used for other public health interventions."

In general, I agree with this assessment though I note that a cost-benefit analysis of testing strategies vis a vis other public health interventions is beyond the scope to this manuscript. However, in my view this discussion of "how useful" screening of unvaccinated (or the entire population) could be made more quantitative and comparable with little extra effort as follows.

In terms of interventions, a popular model is the swiss cheese model, i.e. the fact that multiple imperfect protection layers might provide sufficient protection once they are stacked upon each other. In that regard I find the research question of whether epidemic control can be achieved by testing only (as it is done in the draft) a bit too narrow. Rather, it would be more informative to compare the infection-reducing effect of testing with other non-pharmaceutical interventions. Those are typically evaluated in reduction of the effective reproduction number associated with implementing specific NPIs, see e.g., [1,2].

The authors could, for instance, measure an "early R_{eff} " value in their simulations (or a closely related parameter that can be more easily extracted from the model) for different testing strategies and degrees of compliance. The scenario would be that one observed the onset of a wave and the question is how much testing X% of the unvaccinated or Y% of the vaccinated population reduces R_{eff} in the first couple of days of the outbreak. These percent reductions could be compared with the effectiveness of other NPIs [1,2] to discuss, e.g., how many people need to be tested to achieve a similar effectiveness as closing gastronomy, night clubs, etc. This could in turn inform other, future cost-benefit analysis of public health interventions or under which conditions it could be beneficial to "switch on" such testing regimes.

The second issue is that despite the clear focus of the paper on unvaccinated-only testing strategies, I would find the main messages clearer and more informative if results would also be reported (in the SI) for testing regardless of vaccination status.

In any case, the conclusions drawn in the discussion and in particular in the second paragraph therein should be reevaluated, made more quantitatively, or clarified in light of the fact that the paper so far only reports results for the effectiveness of unvaccinated-only testing and gives no comparisons to benchmark the effectiveness of this intervention with the effectiveness of other interventions.

References:

[1] <https://www.nature.com/articles/s41562-020-01009-0>

[2] <https://www.nature.com/articles/s41467-021-26013-4>

Reviewer #2:

Remarks to the Author:

This work by Bubar and colleagues uses a compartmental model of SARS-CoV-2 transmission to investigate the role of screening of unvaccinated individuals to limit SARS-CoV-2 transmission under different levels of vaccinations.

The manuscript uses mostly theoretical arguments to explore an important topic. The adopted model is adequate to explore the research question. The obtained findings are not surprising, but they provide nice quantitative insights. The study suffers from two major limitations: i) it draws all the conclusions from the analysis of infections only, and ii) it does not consider the clustering of non-vaccinated individuals (e.g., households entirely composed by non-vaccinated individuals, while others are entirely composed by vaccinated ones).

Considering the two limitations above, statements like "As a consequence, our work broadly suggests that unvaccinated-only testing should either cease (...) or be expanded to include the vaccinated population as well (...)" are definitely oversold and the authors should be much more cautious about their statements. Unless a government aims at a zero-COVID policy, infections are not considered to take decisions; deaths, critical, and severe cases would all represent much better metrics. In fact, the repeated screening of non-vaccinated individuals may serve as a tool to protect other unvaccinated individuals in their network of contacts.

Including clustering would not be an easy task considering the modeling framework adopted in this manuscript. A deeper discussion of this limitation and its implications should be added.

On the other hand, including the number of deaths as a supplemental metric is pretty straightforward (and, assuming that the authors already have all the necessary output, could even be done without re-running the simulations). Indeed, using a different fatality risk factor for vaccinated and unvaccinated individuals would show a drastically different (quantitative) figure about when the burden is led by vaccinated individuals. Regardless of the SARS-CoV-2 lineage, vaccine protection against a fatal outcome was continuously estimated in the 90-95% range and could be used as reference values in this analysis.

Definition of the basic reproduction number: "single index" should be replaced with "typical infector". This is a "linguistic" mistake (which needs to be fixed throughout the manuscript) as the authors themselves use the next generation matrix approach, which is entirely based on the concept of "typical" infector, to calculate the reproduction number.

As properly acknowledged by the authors, they use R_0 to refer to a different concept than the basic reproduction number. This may be rather confusing, especially if the paper is read by a non-expert audience. Perhaps a notation such as R_0^{NPI} or something along those lines could avoid the confusion (see for instance <https://bmcmmedicine.biomedcentral.com/articles/10.1186/s12916-022-02243-1>).

In this simple model (SEIR model with exponential duration of both the latent and infectious period), the mean generation time corresponds to the sum of the latent and infectious period. Therefore, the authors are considering a generation time of $3+6=9$ days. This is extremely long for COVID-19. The generation time mostly affects the timing of the simulations, which is not an outcome of interest here (although it may have a very mild effect on the choice of running the simulations for 270 days). If the authors will need to re-run the simulations, this is something that I recommend fixing. Another option, which does not require re-running the simulations, is to re-define the meaning of time in the model (e.g., if a generation time of 6 days is considered, the final time of the simulation in this new unit of measure will become $270*6/9=180$ days).

The authors refer to “testing” of unvaccinated individuals through the manuscript. However, they are evaluating screening strategies based on testing certain segments of the population. The wording should be amended throughout the manuscript.

Reviewer #3:

Remarks to the Author:

The authors did a tremendous amount of analyses examining the impacts of testing strategies on the spread of an infectious disease, incorporating the effects of varying the initial amount of prior immunity and vaccination rates. They developed a deterministic, compartmental model following a typical SEIR scheme and tracking the transmission and infection of individuals who are classified by their vaccination and prior infection status.

The paper first analyzes how the proportion of breakthrough infections and drivers of transmission shift when initial conditions of vaccination rates and prior immunity change. The authors find transition points relating to the amount of vaccination and immunity where the vaccinated community becomes the predominant driver of transmission. Additionally, they test the sensitivity of the transition points under different assumptions of vaccine effectiveness. The authors then explore the effect of different testing strategies on only the unvaccinated population and find 3 parameter regimes that can help evaluate the impact an unvaccinated testing program will have. When vaccination rates are high, testing of only the unvaccinated population can be ineffective at controlling the epidemic.

I think this paper is a nice contribution to the field of epidemic modeling and can be used to explain the high proportion of breakthrough infections observed in some highly vaccinated communities during this omicron wave. It can contribute to the scientific conversation around the best testing strategies in populations that are heterogeneous with respect to the amount of existing immunity, vaccination etc. The explanation of the model, its purpose, and its limitations are described clearly and the results of the analyses are highly detailed. I also appreciate the authors' commitment to open-source coding practices.

I have a couple of minor comments that I've listed below:

- I think there are a couple of typos in Figure 4. The legend says that the a, b, and c show the results of populations with 25%, 75%, and 95% vaccination rates, but the labels of the regions are off by a letter. Region I should correspond to (a), Region II to (b)... etc.
- For figure 6 it might be beneficial to add an example of a prevalence curve for a parameter sample similar to what is shown in figure 1b. While the blue box of 6a is clear in showing that testing does not reduce infections in this scenario, it might help the reader to see the actual infectious curves of the vaccinated and unvaccinated populations
- Small typo in the legend of S5. Figure 7 shows $R_0=4$ right?

Dear Editors and Reviewers,

We thank you for the close reading of our manuscript and for the suggestions for its improvement. While our detailed responses are in the pages that follow, we provide an overview of the major changes here.

First, pandemic priorities have shifted in the time since our initial submission during the early days of the omicron wave. For better or for worse, policymakers worldwide are now focused less on infections and cases, and more on hospitalization—an observation emphasized in the reviews below. To address this reality and your suggestions, we now consider the protective effects of vaccination and prior infection on the progression from infection to hospitalization. As can be seen in new figures and supplementary materials, studying hospitalizations as an endpoint of unvaccinated-only testing programs does not markedly affect the study's conclusions, but broadens its impacts as suggested.

Second, we have shifted the framing and discussion of the paper to consider testing in the context of other NPIs, including the direct modeling and simulation of universal testing of the vaccinated and unvaccinated alike. These changes also broaden the paper, and include both new analyses and altered text.

Detailed changes are listed below, and are highlighted in blue text in the resubmission. We thank you for the consideration, and thank you again for the opportunity to submit this revised manuscript, which we believe has been much improved by the reviews.

Sincerely,

Daniel Larremore, on behalf of the authors

Reviewer #1 (Remarks to the Author):

The manuscript "SARS-CoV-2 Transmission and Impacts of Unvaccinated-Only Testing in Populations of Mixed Vaccination Status" by Bubar et al. presents a modelling study to assess the effectiveness of testing-for-mitigation for different SARS-CoV-2 virus variants and degrees/types of immunization in the population. This is a timely and important contribution that provides valuable input to inform discussions regarding the design of testing strategies to ensure a safer transition to endemicity, as less and less people remain in the population that are immunologically naive.

The paper is well-written and presented, literature review is adequate and methods and results are described very thoroughly. I just have a couple of minor comments regarding the interpretation of the results that the authors might want to consider.

We thank the reviewer for these encouraging comments, and address the suggestions in detail below.

One of the key sections of the paper is in my view the second paragraph in the discussion. The authors write: "Our analysis of plausible omicron variant transmission parameters in particular (Fig. 6) strongly suggests that weekly unvaccinated-only testing is insufficient, and that universal testing programs with high frequency and compliance are required to control transmission in highly vaccinated communities. However, since omicron infections are generally not severe in vaccinated individuals, the resources required to test entire well vaccinated communities may be better used for other public health interventions."

In general, I agree with this assessment though I note that a cost-benefit analysis of testing strategies vis a vis other public health interventions is beyond the scope to this manuscript. However, in my view this discussion of "how useful" screening of unvaccinated (or the entire population) could be made more quantitative and comparable with little extra effort as follows.

In terms of interventions, a popular model is the swiss cheese model, i.e. the fact that multiple imperfect protection layers might provide sufficient protection once they are stacked upon each other. In that regard I find the research question of whether epidemic control can be achieved by testing only (as it is done in the draft) a bit too narrow. Rather, it would be more informative to compare the infection-reducing effect of testing with other non-pharmaceutical interventions. Those are typically evaluated in reduction of the effective reproduction number associated with implementing specific NPIs, see e.g., [1,2].

The authors could, for instance, measure an "early R_{eff} " value in their simulations (or a closely related parameter that can be more easily extracted from the model) for different testing strategies and degrees of compliance. The scenario would be that one observed the onset of a wave and the question is how much testing $X\%$ of the unvaccinated or $Y\%$ of the vaccinated population reduces R_{eff} in the first couple of days of the outbreak. These percent reductions could be compared with the effectiveness of other NPIs [1,2] to discuss, e.g., how many people need to be tested to achieve a similar effectiveness as closing gastronomy, night clubs, etc. This could in turn inform other, future cost-benefit analysis of public health interventions or under which conditions it could be beneficial to "switch on" such testing regimes.

We thank the reviewer for these suggestions, which tie the paper more closely with the suite of countermeasures available to policymakers, a point which we now include in the introduction and discussion. In doing so, we now emphasize and clarify in the revised manuscript that the R_{eff} estimation which the reviewer suggests can be computed analytically using the formulas in the Methods — including testing of everyone, or just the unvaccinated. We thank the reviewer for suggesting key references as well, which are now cited.

The second issue is that despite the clear focus of the paper on unvaccinated-only testing strategies, I would find the main messages clearer and more informative if results would also be reported (in the SI) for testing regardless of vaccination status.

We now report these results in the SI via multiple new figures, and include the appropriate formula for computing R_{eff} as well. One interesting finding, to which we call the reviewer's attention, is that this idea of an effective testing envelope, bounded by $R=1$ (without testing) and $R=1$ (with testing) works well in both the unvaccinated-only *and* universal testing scenarios.

In any case, the conclusions drawn in the discussion and in particular in the second paragraph therein should be reevaluated, made more quantitatively, or clarified in light of the fact that the paper so far only reports results for the effectiveness of unvaccinated-only testing and gives no comparisons to benchmark the effectiveness of this intervention with the effectiveness of other interventions.

We agree, and have updated the discussion and introduction to better frame the issue and our findings, including specific numerical comparisons of reductions in R in the discussion.

References:

[1] <https://www.nature.com/articles/s41562-020-01009-0>

[2] <https://www.nature.com/articles/s41467-021-26013-4>

Reviewer #2 (Remarks to the Author):

This work by Bubar and colleagues uses a compartmental model of SARS-CoV-2 transmission to investigate the role of screening of unvaccinated individuals to limit SARS-CoV-2 transmission under different levels of vaccinations.

The manuscript uses mostly theoretical arguments to explore an important topic. The adopted model is adequate to explore the research question. The obtained findings are not surprising, but they provide nice quantitative insights. The study suffers from two major limitations: i) it draws all the conclusions from the analysis of infections only, and ii) it does not consider the clustering of non-vaccinated individuals (e.g., households entirely composed by non-vaccinated individuals, while others are entirely composed by vaccinated ones).

We thank the reviewer for this encouraging assessment.

Considering the two limitations above, statements like “As a consequence, our work broadly suggests that unvaccinated-only testing should either cease (...) or be expanded to include the vaccinated population as well (...)” are definitely oversold and the authors should be much more cautious about their statements. Unless a government aims at a zero-COVID policy, infections are not considered to take decisions; deaths, critical, and severe cases would all represent much better metrics. In fact, the repeated screening of non-vaccinated individuals may serve as a tool to protect other unvaccinated individuals in their network of contacts.

We thank the reviewer for this assessment, and have overhauled the language of our conclusions to be more cautious, as suggested. More concrete comments follow.

Including clustering would not be an easy task considering the modeling framework adopted in this manuscript. A deeper discussion of this limitation and its implications should be added.

We agree, and have expanded our treatment of this issue in the Discussion. Of potential interest to the reviewer: in an early iteration of this project, we worked on a version of this model that included homophily by vaccination status, using the present modeling framework. Unfortunately, as the reviewer has predicted, we found that it was difficult to simultaneously (1) freely adjust the intensity of homophily, (2) freely adjust the vaccinated fraction, and (3) keep overall per-individual contact rates constant, without rather clumsy and ad hoc corrections that reined in homophily when very few or very many people were vaccinated. We concluded that a network-level model would be more appropriate to investigate homophily's effects.

On the other hand, including the number of deaths as a supplemental metric is pretty straightforward (and, assuming that the authors already have all the necessary output, could even be done without re-running the simulations). Indeed, using a different fatality risk factor for vaccinated and unvaccinated individuals would show a drastically different (quantitative) figure about when the burden is led by vaccinated individuals. Regardless of the SARS-CoV-2 lineage, vaccine protection against a fatal outcome was continuously estimated in the 90-95% range and could be used as reference values in this analysis.

This is an excellent suggestion, which we have modified only slightly to focus on hospitalizations rather than mortality. We made this choice for two reasons. First, we believe that estimating hospitalizations will be roughly of equivalent interest and relevance to readers. Second, we hoped to include protective effects of prior infection and vaccination, for both delta and omicron, and found far more estimates of VE for hospitalization endpoints rather than mortality. We hope that the reviewer finds that our approach follows the spirit of the supplemental suggestion.

We also found the suggested treatment of deaths (now hospitalizations) to be valuable to include in the main text, as well as the supplement. As a result, there are now new figure panels in the main text and new figures in the supplement.

Definition of the basic reproduction number: “single index” should be replaced with “typical infector”. This is a “linguistic” mistake (which needs to be fixed throughout the manuscript) as the authors themselves use the next generation matrix approach, which is entirely based on the concept of “typical” infector, to calculate the reproduction number.

We agree, and have fixed this error.

As properly acknowledged by the authors, they use R_0 to refer to a different concept than the basic reproduction number. This may be rather confusing, especially if the paper is read by a non-expert audience. Perhaps a notation such as R_0^{NPIs} or something along those lines could avoid the confusion (see for instance <https://bmcmmedicine.biomedcentral.com/articles/10.1186/s12916-022-02243-1>).

We agree, and have changed notation throughout the manuscript accordingly.

In this simple model (SEIR model with exponential duration of both the latent and infectious period), the mean generation time corresponds to the sum of the latent and infectious period. Therefore, the authors are considering a generation time of $3+6=9$ days. This is extremely long for COVID-19. The generation time mostly affects the timing of the simulations, which is not an outcome of interest here (although it may have a very mild effect on the choice of running the simulations for 270 days). If the authors will need to re-run the simulations, this is something that I recommend fixing. Another option, which does not require re-running the simulations, is to re-define the meaning of time in the model (e.g., if a generation time of 6 days is considered, the final time of the simulation in this new unit of measure will become $270*6/9=180$ days).

We thank the reviewer for surfacing this issue, and indeed, agree that this would essentially result in altering the timescale, provided that both typical latent and infectious durations are proportionally decreased. In a review of the literature to help identify parameters and address this issue, we observed two things.

First, direct observations of serial interval or generation time were generally in tension with individually measured latent or infectious periods (to the extent that the latter were estimated directly at all). While latent period, infectious period, and generation time have all been estimated, they don't directly add up in practice, hinting at model misspecification. As a result, our individual parameter choices (3d, 6d) are individually supported by the literature, yet together, the generation time (9d) is not. This is due to the modeling assumption that individuals in the *Infectious* compartment are equally infectious throughout, an assumption violated by analyses of individual-level viral load trajectories which typically show far higher viral loads during the early days -vs- later days of infection. Second, we also observed that this modeling assumption and the tension/mismatch it creates—which the reviewer has highlighted—are seldom discussed among model limitations.

In our revised manuscript, we now incorporate the issue into the Discussion explicitly, calling attention to the implicit assumptions of this ubiquitous class of ODE models, while maintaining the connection with the existing literature.

We realize that this may be viewed as an unsatisfactory approach to addressing the reviewer's comment. An alternative, which could be pursued if deemed necessary, would be to create two *Infectious* compartments, both of which (in series) continue to match total observed infectious periods, but the first of which is more infectious than the second, thus shifting infections earlier, and decreasing the generation time. At present, we believe that this would increase the complexity of the model without altering the study's conclusions, but bring this up as a possible approach if the reviewers and editor prefer the more complex model in the context of the other revisions.

The authors refer to “testing” of unvaccinated individuals through the manuscript. However, they are evaluating screening strategies based on testing certain segments of the population. The wording should be amended throughout the manuscript.

We have audited and adjusted our use of this wording throughout. In most cases, “screening” simply replaces “testing” while in others, we use both words to specify that this is test-based screening. We hope that our changes have improved the precision of the manuscript.

Reviewer #3 (Remarks to the Author):

The authors did a tremendous amount of analyses examining the impacts of testing strategies on the spread of an infectious disease, incorporating the effects of varying the initial amount of prior immunity and vaccination rates. They developed a deterministic, compartmental model following a typical SEIR scheme and tracking the transmission and infection of individuals who are classified by their vaccination and prior infection status.

The paper first analyzes how the proportion of breakthrough infections and drivers of transmission shift when initial conditions of vaccination rates and prior immunity change. The authors find transition points relating to the amount of vaccination and immunity where the vaccinated community becomes the predominant driver of transmission. Additionally, they test the sensitivity of the transition points under different assumptions of vaccine effectiveness. The authors then explore the effect of different testing strategies on only the unvaccinated population and find 3 parameter regimes that can help evaluate the impact an unvaccinated testing program will have. When vaccination rates are high, testing of only the unvaccinated population can be ineffective at controlling the epidemic.

I think this paper is a nice contribution to the field of epidemic modeling and can be used to explain the high proportion of breakthrough infections observed in some highly vaccinated communities during this omicron wave. It can contribute to the scientific conversation around the best testing strategies in populations that are heterogeneous with respect to the amount of existing immunity, vaccination etc. The explanation of the model, its purpose, and its limitations are described clearly and the results of the analyses are highly detailed. I also appreciate the authors' commitment to open-source coding practices.

We thank the reviewer for this summary and positive assessment of the manuscript (and open-source standards—an area of personal interest and commitment for many in our group!).

I have a couple of minor comments that I've listed below:

-I think there are a couple of typos in Figure 4. The legend says that the a, b, and c show the results of populations with 25%, 75%, and 95% vaccination rates, but the labels of the regions are off by a letter. Region I should correspond to (a), Region II to (b)... etc.

-For figure 6 it might be beneficial to add an example of a prevalence curve for a parameter sample similar to what is shown in figure 1b. While the blue box of 6a is clear in showing that testing does not reduce infections in this scenario, it might help the reader to see the actual infectious curves of the vaccinated and unvaccinated populations

-Small typo in the legend of S5. Figure 7 shows $R_0=4$ right?

These issues have now been fixed.

Reviewers' Comments:

Reviewer #2:

Remarks to the Author:

My previous comments have been adequately addressed. I have just a couple of very minor comments that can be easily addressed by the authors.

Second paragraph of the Discussion. I suggest mentioning here that herd immunity may not be achievable in a context of waning of immunity and/or continuous emergence of new variants.

Third paragraph of the Discussion. It should be stressed that there is huge uncertainty surrounding quantitative estimates of the effect of NPIs in reducing R_t .

Reviewer #2 (Remarks to the Author):

My previous comments have been adequately addressed. I have just a couple of very minor comments that can be easily addressed by the authors.

Second paragraph of the Discussion. I suggest mentioning here that herd immunity may not be achievable in a context of waning of immunity and/or continuous emergence of new variants.

Third paragraph of the Discussion. It should be stressed that there is huge uncertainty surrounding quantitative estimates of the effect of NPIs in reducing R_t .

We have added two sentences to the discussion in paragraphs two and three which address these two suggestions, respectively. We thank Reviewer 2 not only for the continued constructive feedback, but also for the swiftness of these additional comments.